# Forward genetics in *C. elegans* reveals genetic adaptations to polyunsaturated fatty acid deficiency

**Delaney Kaper[1], Uroš Radović[1], Per-Olof Bergh[2], August Qvist[1], Marcus Henricsson[2], Jan Borén[2], Marc Pilon[1]***

[1]Department of Chemistry and Molecular Biology, University of Gothenburg, Gothenburg, Sweden; [2]Department of Molecular and Clinical Medicine/Wallenberg Laboratory, Institute of Medicine, University of Gothenburg, Gothenburg, Sweden

## eLife Assessment

This **fundamental** study investigates the role of polyunsaturated fatty acids (PUFAs) in physiology and membrane biology, using a unique model to perform a thorough genetic screen that demonstrates that PUFA synthesis defects cannot be compensated for by mutations in other pathways. These findings are supported by **compelling** evidence from a high quality genetic screen, functional validation of their hits, and lipid analyses. This study will appeal to researchers in membrane biology, lipid metabolism, and *C. elegans* genetics.

*****For correspondence:
marc.pilon@cmb.gu.se

**Abstract** Polyunsaturated fatty acids (PUFAs) are essential for mammalian health and function as membrane fluidizers and precursors for signaling lipids, though the primary essential function of PUFAs within organisms has not been established. Unlike mammals who cannot endogenously synthesize PUFAs, *C. elegans* can de novo synthesize PUFAs starting with the Δ12 desaturase FAT-2, which introduces a second double bond to monounsaturated fatty acids to generate the PUFA linoleic acid. FAT-2 desaturation is essential for *C. elegans* survival since *fat-2* null mutants are non-viable; the near-null *fat-2(wa17)* allele synthesizes only small amounts of PUFAs and produces extremely sick worms. Using fluorescence recovery after photobleaching (FRAP), we found that the *fat-2(wa17)* mutant has rigid membranes and can be efficiently rescued by dietarily providing various PUFAs, but not by fluidizing treatments or mutations. With the aim of identifying mechanisms that compensate for PUFA-deficiency, we performed a forward genetics screen to isolate novel *fat-2(wa17)* suppressors and identified four internal mutations within *fat-2* and six mutations within the HIF-1 pathway. The suppressors increase PUFA levels in *fat-2(wa17)* mutant worms and additionally suppress the activation of the *daf-16*, UPR$^{er}$ and UPR$^{mt}$ stress response pathways that are active in *fat-2(wa17)* worms. We hypothesize that the six HIF-1 pathway mutations, found in *egl-9*, *ftn-2*, and *hif-1*, all converge on raising Fe$^{2+}$ levels and in this way boost desaturase activity, including that of the *fat-2(wa17)* allele. We conclude that PUFAs cannot be genetically replaced and that the only genetic mechanism that can alleviate PUFA-deficiency do so by increasing PUFA levels.

## Introduction

The fluidity of cellular membranes is heavily influenced by the saturation level of the phospholipids composing the membrane. Phospholipids containing saturated fatty acid (SFA) tails are more tightly packed and, therefore, form more rigid membranes, while phospholipids with an abundance of unsaturated fatty acids (UFAs) are more loosely packed and result in fluid membranes (*Barelli and Antonny,*

*2016*; *Antonny et al., 2015*). PUFAs themselves can also affect many different cellular processes and are precursors to both anti- and pro-inflammatory PUFA-derived signaling molecules called eicosanoids (*Bazinet and Layé, 2014*; *Simopoulos, 1999*; *James et al., 2000*; *Bell et al., 1986*). Imbalances between SFAs and PUFAs are associated with chronic diseases including coronary heart disease, diabetes, hypertension, and renal disease (*Simopoulos, 1999*).

*C. elegans* PAQR-2, and its mammalian ortholog AdipoR2, promote the production and incorporation of PUFAs into phospholipids to restore membrane homeostasis (*Ruiz et al., 2023*). Whether this is the primary function of PUFAs in cells or in organismal physiology is still not resolved. In particular, no unbiased forward genetic screens for suppressors of PUFA deficiency have been reported. Mammals are not able to endogenously synthesize PUFAs and must obtain omega-3 and omega-6 PUFAs from the diet, a fact known since 1930; *Murff and Edwards, 2014*; *Burr and Burr, 1930*; linoleic acid (LA, 18:2n6) and alpha-linolenic acid (ALA, 18:3n3) must be dietarily supplied and can be further desaturated and elongated into ≥20-carbon PUFAs used structurally or as precursors of signaling molecules (*Watts and Browse, 2002*). An exception to this exists during severe essential fatty acid deficiency when mammals can synthesize mead acid (20:3n9), though this is not a common occurrence (*Ichi et al., 2014*). In contrast, *C. elegans* expresses many desaturases and elongases that can convert dietary or de novo synthesized SFAs into a wide range of PUFAs: Δ9 desaturases are responsible

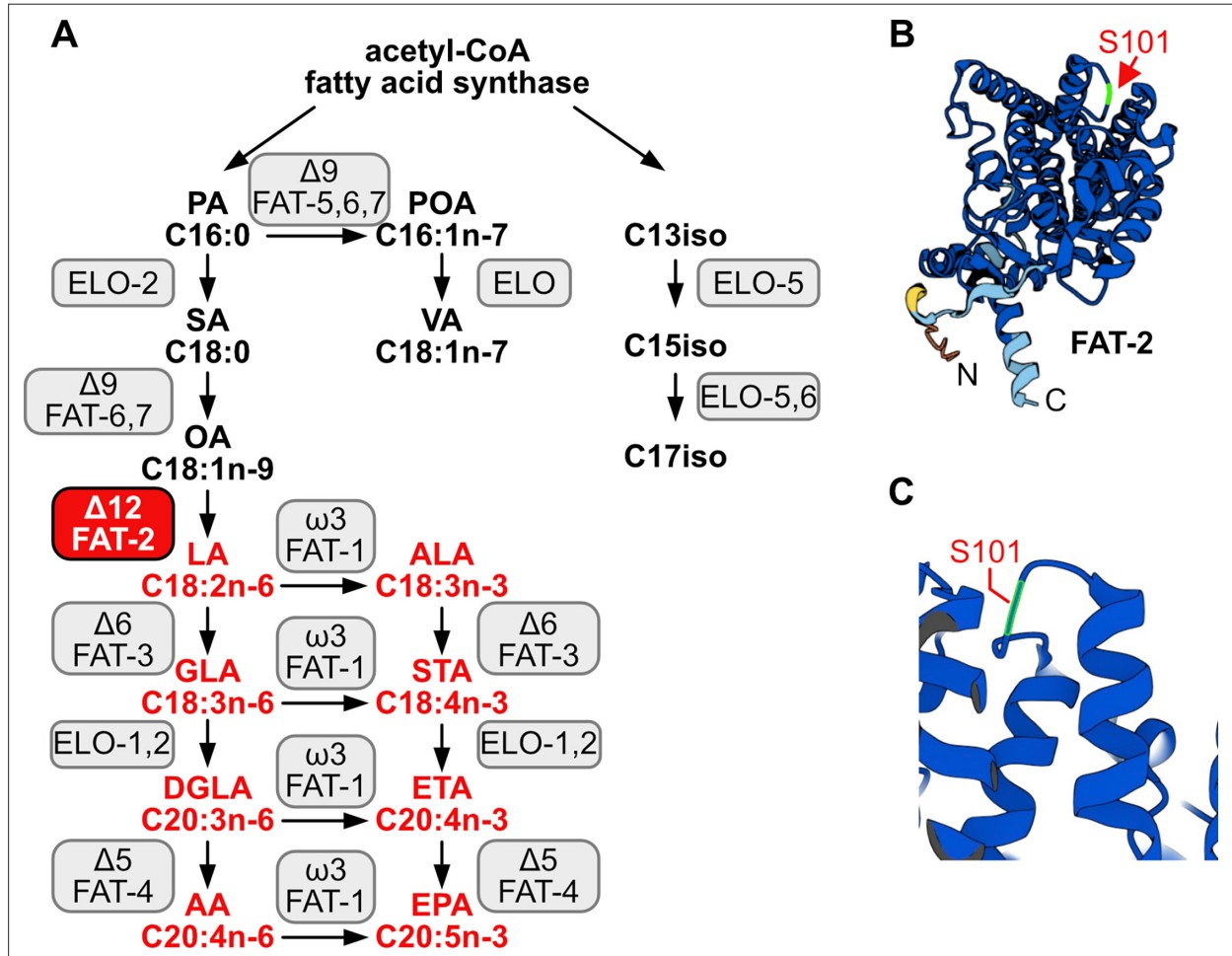

**Figure 1.** *C.elegans* fatty acid synthesis pathway and FAT-2 desaturase. (**A**) Simplified pathway of fatty acid synthesis and desaturation in *C. elegans*. Boxes indicate the name of the enzymes, with the FAT-2 desaturase being indicated in a red box. Fatty acids whose synthesis is dependent on FAT-2 are indicated in red. Fatty acid abbreviations are as follow: palmitic acid (PA), palmitoleic acid (POA), vaccenic acid (VA), stearic acid (SA), oleic acid (OA), linoleic acid (LA), alpha-linolenic acid (ALA), gamma-linolenic acid (GLA), stearidonic acid (STA), dihomo-gamma-linolenic acid (DGLA), eicosatetraenoic acid (ETA), arachidonic acid (AA), and eicosapentaenoic acid (EPA). (**B**) AlphaFold2 predicted the FAT-2 structure with the serine at position 101 indicated with a red arrow. (**C**) Same structure as in B, zoomed in and angled to show that the S101 position that is mutated to phenylalanine in the *fat-2(wa17)* allele lies in a loop connecting two alpha helices.

for converting SFAs into monounsaturated fatty acids (MUFAs) by adding a first double bond, a Δ12 desaturase adds an additional double bond to transform MUFAs into LA, a PUFA with two double bonds, and Δ5 and Δ6 desaturases introduce additional double bonds to produce PUFAs with three, four, or five double bonds (*Nakamura and Nara, 2004*; *Guillou et al., 2010*; *Zhang et al., 2016*). To better understand the essential roles of PUFAs in cells and whole organisms, we leveraged a mutant allele of the *C. elegans* Δ12 desaturase FAT-2, whose function is to convert oleic acid (OA, 18:1n9) into linoleic acid (LA, 18:2n6) (*Peyou-Ndi et al., 2000*; *Figure 1A*). Because this is a critical step for PUFA production, worms devoid of FAT-2 activity (i.e. *fat-2(-)* null mutants) are not able to synthesize any PUFAs and are not viable. In contrast, the *fat-2(wa17)* allele produces a partially functional protein bearing a S101F substitution (*Figure 1B–C*): mutants homozygous for this allele produce <10% of the normal levels of PUFAs and are extremely slow growing and sickly, but, crucially, are viable (*Watts and Browse, 2002*).

Although there is no mammalian homolog of FAT-2 (*Zhou et al., 2011*), the *fat-2(wa17)* mutant can still serve as a useful genetic model to reveal evolutionarily conserved roles of PUFAs in organisms, and to help identify mechanisms that can compensate for PUFA deficiency. Here, we began by characterizing the *fat-2(wa17)* mutant in terms of its ability to be rescued with dietary PUFAs or with mutations previously identified as suppressors of *paqr-2(tm3410)* mutant phenotypes that are attributed to membrane rigidity. We then performed an exhaustive forward genetic screen for *fat-2(wa17)* suppressors in the hope that the critical functions of PUFAs could be discovered and to identify mechanisms that allow cells/organisms to cope with PUFA deficiencies. This screen yielded ten *fat-2(wa17)* suppressor alleles that fell into two groups: mutations within *fat-2* itself, and mutations in the *hif-1* pathway that converge on inhibition of *ftn-2* expression.

## Results

### Characterization of the *fat-2(wa17)* mutant

The severe growth defect of *fat-2(wa17)* mutants can be suppressed by the wild-type *fat-2(+)* allele carried on an extrachromosomal array, confirming that this growth defect is due to reduced *fat-2* activity (*Figure 2A*). As expected, given their low amounts of PUFAs, fluorescence recovery after photobleaching (FRAP) shows that the membranes of intestinal cells in the *fat-2(wa17)* mutant are excessively rigid and indeed appeared as rigid as those of the *paqr-2(tm3410)* mutant characterized by an excess of SFAs in its phospholipids (*Svensk et al., 2013*; *Svensk et al., 2016*; *Figure 2B–C*).

Also, as expected, low doses of dietary linoleic acid (LA, 18:2n6) fully rescued the *fat-2(wa17)* growth defect (*Figure 2D*). This rescue by LA is transient and does not last when the following generation is transferred back to NGM plates (*Figure 2E*), which is consistent with the rapid turnover of fatty acids in *C. elegans* (*Dancy et al., 2015*). Eicosapentaenoic acid (EPA, 20:5n3) is the longest and most unsaturated PUFA produced in *C. elegans* (*Figure 1A*) and is also able to rescue the *fat-2(wa17)* mutant, albeit requiring higher concentrations than LA (*Figure 2F*). Surprisingly, docosahexaenoic acid (DHA, 22:6n3), which is not produced by *C. elegans*, is also able to rescue the *fat-2(wa17)* mutant (*Figure 2G*). Temperature has a direct effect on membrane fluidity: given a constant phospholipid composition, lower temperatures cause rigidification while higher temperatures promote fluidity (*Tanaka et al., 1996*). We found that the *fat-2(wa17)* mutant is growth-arrested when cultivated at membrane-rigidifying 15 °C, and conversely shows improved growth at 25 °C, which again is similar to earlier findings with the *paqr-2(tm3410)* mutant (*Devkota et al., 2021*; *Figure 2H*). However, cultivating the *fat-2(wa17)* mutant in the presence of the non-ionic detergent NP-40, which improves the growth of the *paqr-2(tm3410)* mutant (*Svensk et al., 2013*), did not suppress the poor growth phenotype of the *fat-2(wa17)* mutant even though it did improve membrane fluidity as measured using FRAP (*Figure 2I–J*). Similarly, supplementing the *fat-2(wa17)* mutant with the MUFA oleic acid (OA, 18:1), which also suppresses *paqr-2(tm3410)* phenotypes (*Svensk et al., 2013*), did not suppress the poor growth phenotype of the *fat-2(wa17)* mutant (*Figure 2K*). Conversely, membrane-rigidifying glucose (which causes a high SFA/UFA ratio in the dietary *E. coli*; *Devkota et al., 2017*) or the SFA palmitic acid (PA, 16:0) did not exacerbate the growth defect of *fat-2(wa17)* (*Figure 2L–M*). The observations that NP-40 and OA do not rescue *fat-2(wa17)*, and that membrane rigidifying conditions (dietary glucose or PA) were not detrimental to *fat-2(wa17)*, suggest that membrane rigidification may not be the main cause of the *fat-2(wa17)* growth defects.

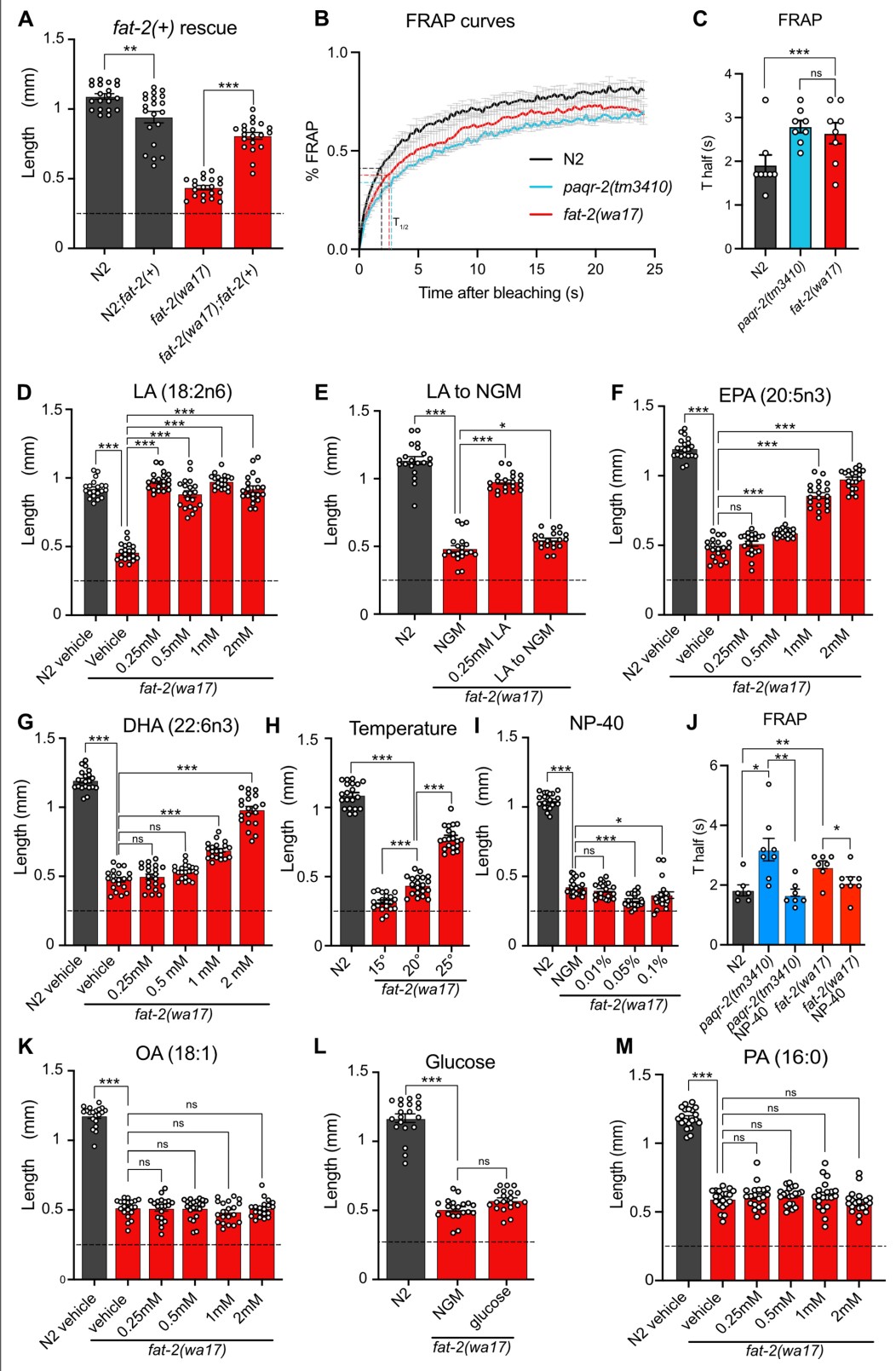

**Figure 2.** Characterization and rescue of *fat-2(wa17)*. (**A**) Introduction of the wild-type *fat-2(+)* allele on an extrachromosomal array rescues the *fat-2(wa17)* growth defect. n=20 for each genotype. (**B–C**) Fluorescence recovery after photobleaching (FRAP) curve and T_half value show that *fat-2(wa17)* has rigid membranes similar to *paqr-2(tm3410)* control. n=8 for each genotype. (**D–I**, **K–M**) The lengths of *fat-2(wa17)* worms grown from L1 stage

*Figure 2 continued on next page*

*Figure 2 continued*

for 72 hr in the indicated conditions; horizontal dashed lines indicate the approximate lengths of the synchronized L1s at the start of the experiments. n=20 for each genotype/condition. (**J**) FRAP $T_{half}$ values show that NP-40 rescues *fat-2(wa17)* rigid membranes similarly to *paqr-2(tm3410)*. From left to right, n=6, 8, 7, 7, 8. Error bars show the standard error of the mean. *p<0.05, **p<0.01, ***p<0.001 indicate significant differences compared to the *fat-2(wa17)* control (ordinary one-way ANOVA with Tukey multiple comparisons test).

Lipidomic analysis of phosphatidylcholines (PCs) and phosphatidylethanolamines (PEs) in the *fat-2(wa17)* mutant confirmed its reduced levels of PUFAs relative to wild-type worms (*Figure 3A*; *Figure 3—figure supplement 1A*), with the largest loss observed in longer PUFAs, namely diho-mo-γ-linolenic acid (DGLA; C20:3), arachidonic acid or eicosatetraenoic acid (AA or ETA, C20:4) and, most strikingly, EPA (C20:5) (*Figure 3B–C*; *Figure 3—figure supplement 1B–C*). Consistent with previous publications (*Watts and Browse, 2002*), the levels of 18:1 fatty acids were greatly increased in the *fat-2(wa17)* mutant. Even though the lipid analysis methods used here are not able to distin-guish between different 18:1 species, a previous study showed that the majority of the 18:1 fatty acids in the *fat-2(wa17)* mutant is actually 18:1n9 (OA) (*Watts and Browse, 2002*) and not 18:1n7 (vaccenic acid) as in most other strains (*Watts and Browse, 2002*; *Hutzell and Krusberg, 1982*); this is because OA is the substrate of FAT-2 and thus accumulates in the mutant. As expected, exogenous addition of LA to these worms resulted in an increase of PUFAs, and in particular, resulted in elevating EPA levels to more than 12% compared to less than 2% of total fatty acids in the PCs of *fat-2(wa17)*. Transferring *fat-2(wa17)* from LA to NGM 6 hr prior to harvesting did not lessen this increase, suggesting minimal LA depletion during this time (*Figure 3A–C*; *Figure 3—figure supplement 1A–C*). Cultivation at 25°C did not result in any significant changes in the *fat-2(wa17)* mutant, indicating that the growth rescue seen at 25°C is not due to increased PUFA levels in the lipidome (*Figure 3A–C*; *Figure 3—figure supplement 1A–C*).

## Testing the effect of *paqr-2(tm3410)* suppressors

The previously characterized *paqr-2(tm3410)* null mutant has excess SFAs within phospholipids and many of the resulting defects, including membrane rigidity, can be suppressed by mutations that activate fatty acid desaturases or promote the incorporation of UFAs into phospholipids (*Svensk et al., 2013*; *Ruiz et al., 2018*; *Ruiz et al., 2019*; *Busayavalasa et al., 2020*). Given the phenotypic similarities between *paqr-2(tm3410)* and *fat-2(wa17)*, such as cold intolerance and rigid membranes, we hypothesized that previously characterized *paqr-2(tm3410)* suppressors may be able to suppress *fat-2(wa17)* as well. However, the *paqr-2(tm3410)* suppressors tested (*mdt-15(et14)*, *nhr-49(et8)*, *fld-1(et46)*, *paqr-1(et52)*, *acs-13(et54)*; *Svensk et al., 2013*; *Ruiz et al., 2018*; *Ruiz et al., 2019*; *Busayav-alasa et al., 2020*) resulted in either no or only slight rescue of *fat-2(wa17)* growth (*Figure 4A–D*). Additionally, Oil Red O staining of *fat-2(wa17)* mutants suggests that they have an excessive lipid content, and this too was not normalized by the tested *paqr-2(tm3410)* suppressors (*Figure 4E–G*). These results suggest that membrane rigidity is at most only a minor cause of the *fat-2(wa17)* defects since fluidizing treatments (NP-40 or OA) or mutations (the tested *paqr-2* suppressors) provide only minimal or no suppression.

## A forward genetics screen for *fat-2(wa17)* suppressors

With the aim of identifying essential roles of PUFAs and molecular mechanisms that can compen-sate for PUFA deficiency, we performed a forward genetic screen for *fat-2(wa17)* suppressors that allow growth to adulthood within 72 hr, as opposed to the ~120 hr needed for the parental strain (*Figure 5A*). Approximately 40,000 EMS-mutagenized haploid genomes were screened, leading to the isolation of ten *fat-2(wa17)* suppressors, which fell into two groups: mutations within the *fat-2* locus itself and mutations within genes of the HIF-1 pathway (*Figure 5B–C*). The *fat-2(wa17)* suppres-sors all reached adulthood within 72 hr (criteria for the screen) and improved the growth of the mutant when assessed by measuring worm length at 72 hr (*Figure 5D–E*). The *hif-1(et69)* mutation was recre-ated by CRISPR-Cas9 within the *fat-2(wa17)* background to confirm its *fat-2(wa17)* suppressor activity (*Figure 5—figure supplement 1A*); multiple independent isolations of essentially the same alleles within *egl-9* (alleles *et60-et62*), *fat-2* (alleles *et63-et66*), *and ftn-2(et67-68)* also serve as confirmation for these loci.

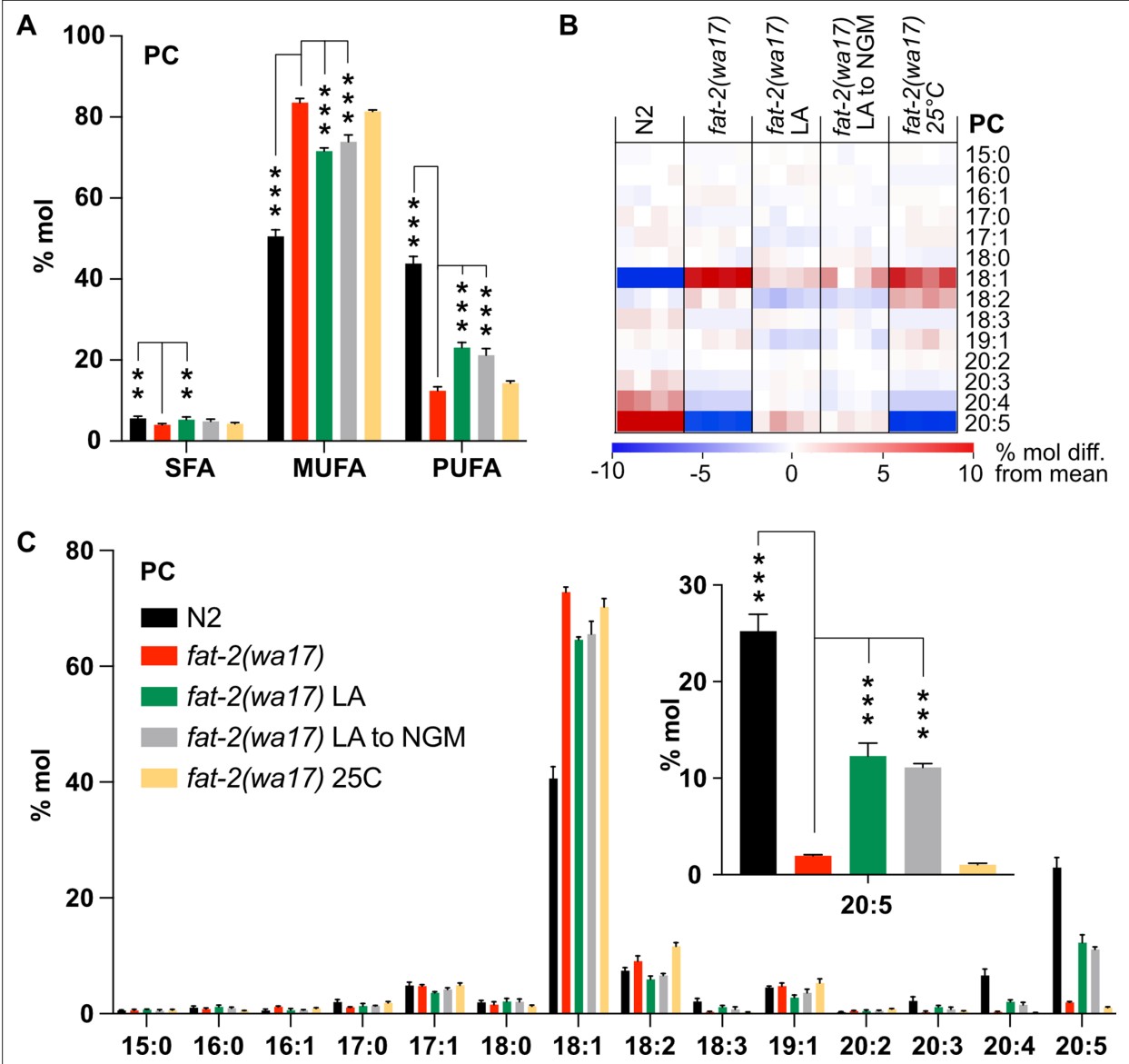

**Figure 3.** Lipidomic analysis of *fat-2(wa17)* mutant. (**A**) Saturated fatty acid (SFA), monounsaturated fatty acid (MUFA), and polyunsaturated fatty acid (PUFA) levels in phosphatidylcholines (PCs) of *fat-2(wa17)* grown in various conditions. Note that cultivation on 2 mM LA boosts PUFA levels. Linoleic acid (LA) to NGM worms were grown on 2 mM LA before being transferred to NGM 6 hr prior to harvesting. (**B**) Heatmap of phosphatidylcholine (PC) species in *fat-2(wa17)* in all conditions. (**C**) Levels of individual FA species in PCs for all conditions. The inset shows that levels of 20:5 FA are increased by providing *fat-2(wa17)* with linoleic acid. n=4 populations for each genotype/condition. For A and C (inset), *p<0.05, **p<0.01, ***p<0.001 indicate significant differences compared to the *fat-2(wa17)* control and using one-way ANOVA followed by a Dunnett's multiple comparison test.

The online version of this article includes the following source data and figure supplement(s) for figure 3:

**Source data 1.** Data from the targeted lipidomics analysis, related to *Figures 3 and 7*, *Figure 3—figure supplement 1*, *Figure 7—figure supplement 1*.

**Figure supplement 1.** Lipidomics analysis of phosphatidylethanolamines (PEs) in *fat-2(wa17)* in various cultivation conditions.

Three of the four *fat-2* intragenic alleles (*et64-et66*) carried a substitution of serine to leucine at position 99 (S99L), only two amino acids away from the S101F mutation in *fat-2(wa17)*; the fourth, *et63*, is a missense mutation substituting valine with methionine at position 25 (V25M; *Figure 5B*). These four internal *fat-2* alleles likely compensate structurally for the S101F mutation in *fat-2(wa17)* and thus improve its activity.

Three independent *fat-2(wa17)* suppressor mutations were found to affect the same arginine at position 557 of EGL-9 (*et60 and et61* resulted in a R557H missense while *et62* caused a R557C

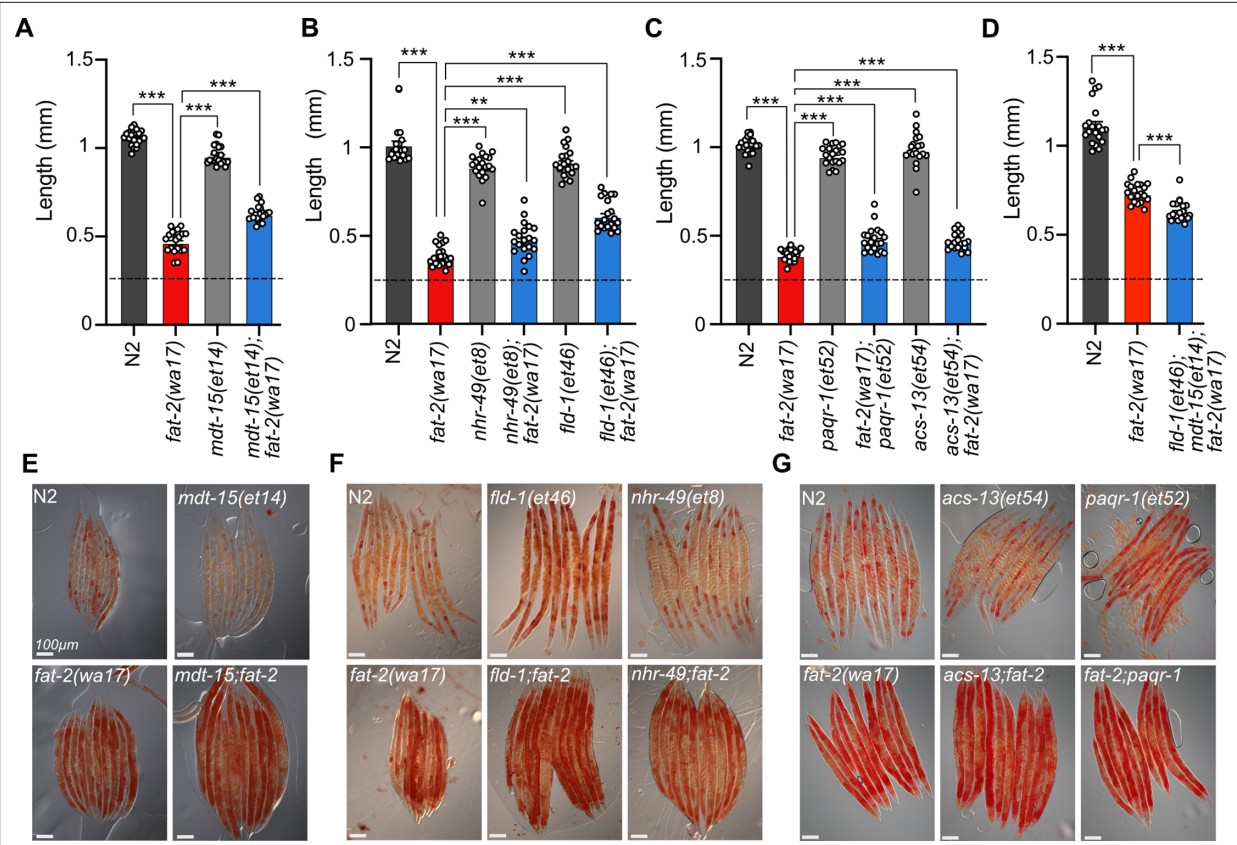

**Figure 4.** Membrane fluidizing mutations partially rescue *fat-2(wa17)*. (A–D) Fluidizing *paqr-2(tm3410)* suppressor mutations only slightly rescue *fat-2(wa17)* growth. Dashed horizontal lines indicate the approximate length of L1s at the start of the experiments; length was measured 72 hr post-synchronization. n=20 for each genotype. Error bars show the standard error of the mean. *p<0.05, **p<0.01, ***p<0.001 indicate significant differences compared to the *fat-2(wa17)* control (ordinary one-way ANOVA with Tukey multiple comparisons test). (E–G) Oil Red O staining of day 1 adults shows that the high lipid abundance in *fat-2(wa17)* is not suppressed by *paqr-2(tm3410)* fluidizing mutations.

missense; *Figure 5B*). EGL-9 is a proline hydroxylase (PHD) that interacts via its R557 with $Fe^{2+}$/2-oxoglutarate, a required cofactor for its oxygenase activity (*Epstein et al., 2001*). EGL-9 regulates the response to iron depletion and hypoxia: in the presence of sufficient $Fe^{2+}$ and oxygen, EGL-9 can hydroxylate HIF-1 (hypoxia-inducible factor 1), leading to its ubiquitination and degradation. When either $Fe^{2+}$ or oxygen are unavailable, HIF-1 is stable and can bind DNA to regulate adaptive transcriptional responses (*Epstein et al., 2001*; *Bracken et al., 2006*; *Huang et al., 1996*). It is surprising that all three *fat-2(wa17)* suppressor alleles affected precisely the same amino acid within EGL-9, and we surmise that a special property is conferred to EGL-9 by this specific mutation. For example, this mutant version of EGL-9 may be unable to inactivate HIF-1 by hydroxylation but still retain other important functions. In agreement with this interpretation, we found that the *egl-9(sa307)* null mutant cannot act as a *fat-2(wa17)* suppressor (*Figure 5F*).

One of the *fat-2(wa17)* suppressors corresponds to a splice acceptor mutation in the fifth intron of HIF-1, which would result in a frameshift after the first 413 amino acids if splicing instead occurs with the following sixth intron splice acceptor site (*Figure 5B*). This *hif-1(et69)* allele is dominant: heterozygosity for *hif-1(et69)/+* provides better *fat-2(wa17)* suppression than *hif-1(et69)/hif-1(et69)* homozygosity (*Figure 5—figure supplement 1B*). This suggests that the *hif-1(et69)* allele is a gain-of-function allele, which may be because the frameshift occurs just after the first of potentially two prolines that are hydroxylated by EGL-9 when oxygen and $Fe^{2+}$ levels are sufficient (*Epstein et al., 2001*). This is also consistent with the observation that the *hif-1(ok2654)* null allele is not a *fat-2(wa17)* suppressor (*Figure 5G*). Usually, hydroxylation of the prolines P400 and P621 causes recruitment of a ubiquitin ligase, leading to HIF-1 degradation (*Epstein et al., 2001*). In the case of the *hif-1(et69)* allele, such regulation is likely impossible, and a constitutive HIF-1 may act as a *fat-2(wa17)* suppressor in several

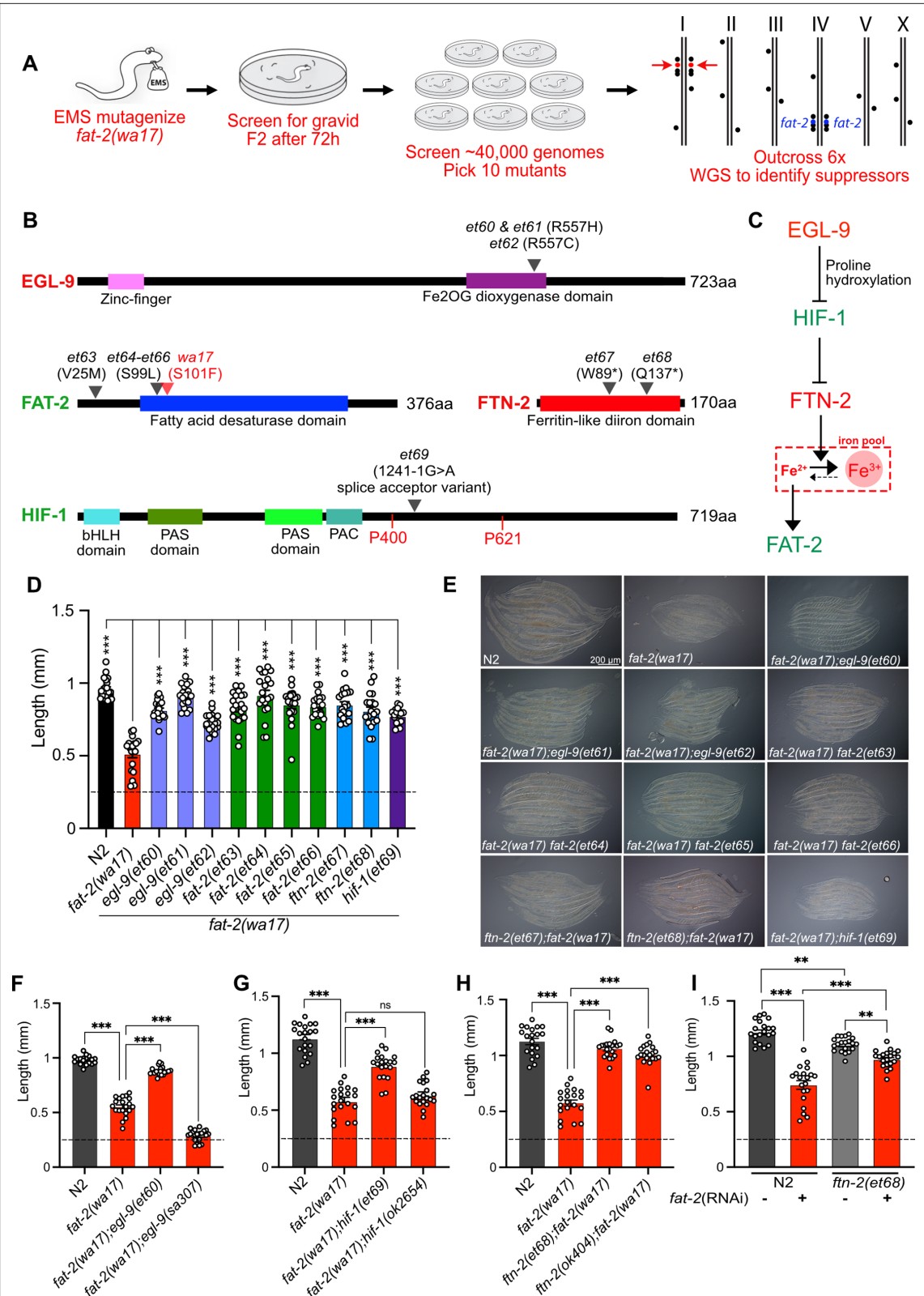

**Figure 5.** A forward genetic screen reveals that *fat-2(wa17)* is suppressed by mutations in the HIF-1 pathway. (**A**) Overview of the forward genetics screen strategy to isolate *fat-2(wa17)* suppressors. (**B**) Identity and position of the *fat-2(wa17)* suppressors as well as the positions of functional domains. Novel mutations are marked by a black triangle with the corresponding allele name and mutation effect; the red triangle in FAT-2 indicates the original *wa17* allele. Gene names in red represent loss- or reduction-of-function mutations; gene names in green represent gain-of-function mutations. (**C**) Proposed

*Figure 5 continued on next page*

Figure 5 continued

pathway of *fat-2(wa17)* suppression by mutations in the HIF-1 pathway. Reduction of EGL-9 constitutively activates HIF-1, and HIF-1 activation inhibits FTN-2. The loss of FTN-2 increases the levels of $Fe^{2+}$, thus boosting FAT-2 desaturase activity. Gain-of-function mutations are labeled in green, loss- or reduction-of-function mutations are labeled in red. (**D**) Length of all *fat-2(wa17)* suppressors measured 72 hr after the L1 stage. (**E**) Representative images of *fat-2(wa17)* suppressors after 72 h of growth. (**F–H**) Null alleles of *egl-9* and *hif-1* do not rescue *fat-2(wa17)*, but the null allele of *ftn-2* does, confirming that *ftn-2(et67)* and *ftn-2(et68)* are loss-of-function alleles. Lengths were measured 72 hr after L1 synchronization. (**I**) *ftn-2(et68)* rescue of *fat-2(RNAi)* worms, confirming that the suppressors are not *wa17* specific. The horizontal dashed line indicates the approximate length of L1s at the start of each experiment. n=20 for each genotype/condition. Error bars show the standard error of the mean. *p<0.05, **p<0.01, ***p<0.001 indicate significant differences compared to the *fat-2(wa17)* control (ordinary one-way ANOVA with Tukey multiple comparisons test).

The online version of this article includes the following figure supplement(s) for figure 5:

**Figure supplement 1.** *fat-2(wa17)* and *fat-2(syb7458)* with suppressors.

ways: promote overexpression of lipid metabolism genes including *fat-2* (*Xie and Roy, 2012*), inhibit fatty acid beta-oxidation (*Huang et al., 2014*; *Papandreou et al., 2006*), which may help PUFAs to reach adequate levels even in the *fat-2(wa17)* mutant, or suppress the expression of the ferritin-encoding *ftn-2*, thus increasing the levels of ferrous ions required for desaturase activity (*Romney et al., 2011*; *Shen et al., 2023*).

Most informatively, the last two *fat-2(wa17)* suppressor mutations introduced premature STOP codons within the *ftn-2* gene (alleles *et67* and *et68*; *Figure 5B*). Additionally, the *ftn-2(ok404)* null allele also acted as a potent *fat-2(wa17)* suppressor (*Figure 5H*), which is consistent with inhibition of *ftn-2* being the key outcome from HIF-1 pathway activation. *C. elegans ftn-2* encodes a ferritin that is expressed in the intestine, muscle and several neurons (*Romney et al., 2008*). The FTN-2 protein is constitutive and 10 X faster as a ferroxidase (oxidising the reactive ferrous $Fe^{2+}$ to the harmless ferric $Fe^{3+}$) than FTN-1, which is an inducible intestine-specific ferritin in *C. elegans* (*Romney et al., 2011*; *Kim et al., 2004*; *Romero et al., 2020*; *Mubarak et al., 2023*; *Cha'on et al., 2007*). Additionally, FTN-2 is the major binder of iron in worms, and *ftn-2* mutants, therefore, contain much less iron than wild-type worms, though the $Fe^{2+}/Fe^{3+}$ ratio is increased among the remaining iron (*James et al., 2015*). This is likely the mechanism by which the *ftn-2(et67)* and *ftn-2(et68)* alleles act as *fat-2(wa17)* suppressors: increasing the availability of ferrous ions is a potent way to activate desaturases (*Shen et al., 2023*) and thus likely increases the activity of the near null *fat-2(wa17)* allele leading to the production of more/sufficient PUFAs. Importantly, the *ftn-2(et68)* allele was also able to suppress the growth defect resulting from *fat-2* knockdown (using RNAi; *Figure 5I*); this shows that ferritin mutations compensate for reduced *fat-2* activity generally rather than suppressing specifically only the *fat-2(wa17)* allele. Additionally, the *ftn-2(et68)* allele was not able to rescue the *fat-2(syb7458)* null allele (*Figure 5—figure supplement 1C*), suggesting that some *fat-2* activity must exist for *ftn-2(et68)* to act upon. Lastly, *ftn-2(et68)* is still a potent *fat-2(wa17)* suppressor when *hif-1* is knocked out (*Figure 5—figure supplement 1D*), suggesting that no other HIF-1-dependent functions are required as long as *ftn-2* is downregulated; this conclusion is supported by the observation that the potency of the *ftn-2(ok404)* null allele to act as a *fat-2(wa17)* suppressor is not increased by including the *hif-1(et69)* allele (compare *Figure 5H* and *Figure 5—figure supplement 1E*). Altogether, the genetic interaction studies suggest that the suppressor mutations in *ftn-2* and *hif-1* are acting via the same mechanism to rescue *fat-2(wa17)* and that *ftn-2* is downstream of *hif-1* in the *fat-2* suppression pathway.

## Effect of *fat-2(wa17)* suppressors on HIF-1 and PUFA levels

The results of the *fat-2(wa17)* suppressor screen support a model where the *egl-9* R557 substitution alleles have an impaired ability to suppress HIF-1, while the gain-of-function *hif-1(et69)* allele constitutively suppresses *ftn-2* expression and the *ftn-2* null alleles are unable to sequester ferrous ions of which elevated levels increase *fat-2* activity (*Figure 5C*). A 3xFLAG-tagged version of the endogenous HIF-1 allowed us to monitor HIF-1 levels in different conditions using Western blots (*Figure 6A–B*). As expected, hypoxia caused elevated levels of HIF-1 in wild-type worms. HIF-1 levels are abnormally low in *fat-2(wa17)* during normoxia but restored to normal levels by the internal *fat-2(et65)* mutation, suggesting that low PUFA levels cause HIF-1 downregulation. Nevertheless, HIF-1 levels are also increased by hypoxia in the *fat-2(wa17)* mutant, indicating that the *hif-1* locus is still responsive to oxygen levels in the *fat-2(wa17)* mutant. As expected, *egl-9(et60)* drastically increases HIF-1 expression in *fat-2(wa17)*, which is consistent with the R557 substitution impairing the ability of EGL-9 to

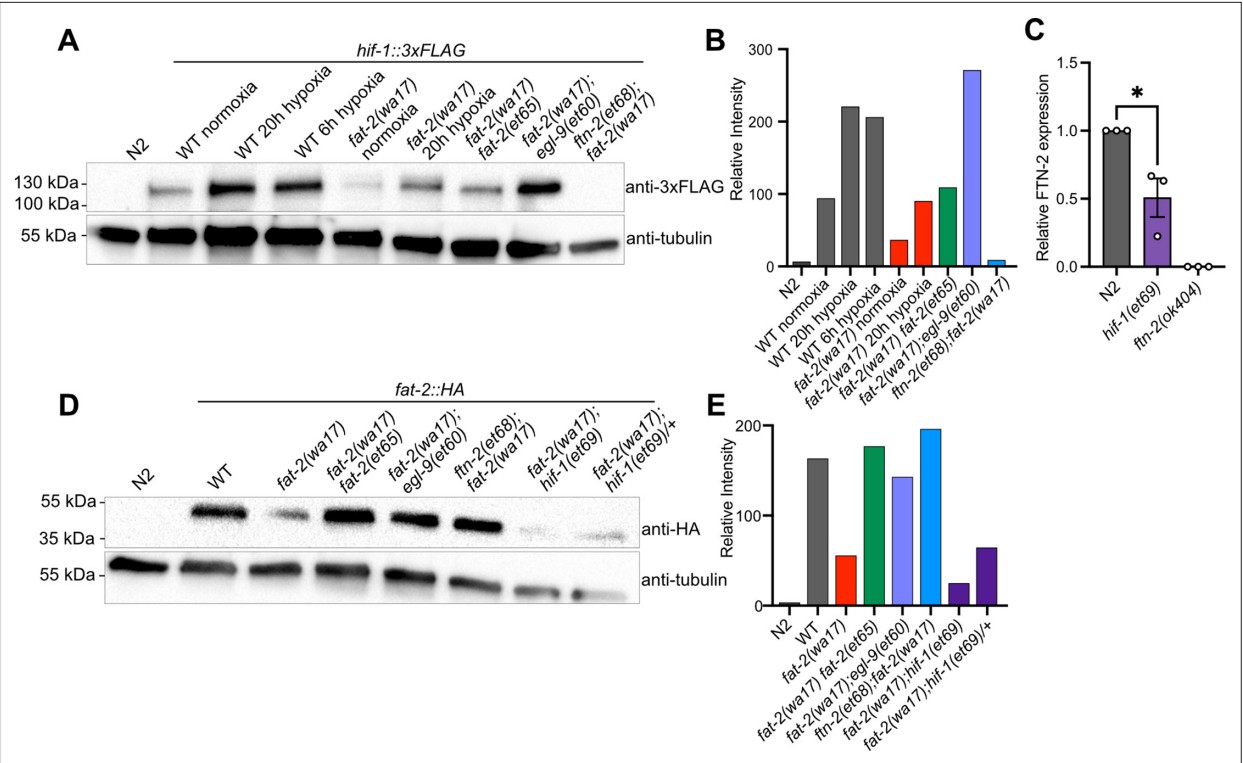

**Figure 6.** *fat-2(wa17)* suppressors belong in the HIF-1 pathway and influence HIF-1 levels. (**A**) Western blot confirming that *hif-1::3xFLAG* levels in *fat-2(wa17)* are increased by *egl-9(et60)*, but not by *ftn-2(et68)*. Hypoxia treatment increases HIF-1 levels in wild-type (WT) and *fat-2(wa17)*, confirming successful protein tagging. (**B**) Quantification of Western blot in **A** showing relative intensity of the HIF-1 signal normalized to that of tubulin. (**C**) mRNA expression of FTN-2, confirming that *hif-1(et69)* reduces FTN-2 levels. n = the mean of 3 independent normalized replicates for each genotype. *p<0.05 (unpaired t-test). (**D**) Western blot confirming that *fat-2::HA* levels *in fat-2(wa17)* are greatly reduced but increased in suppressor strains. (**E**) Quantification of Western blot in D showing relative intensity of the FAT-2 signal normalized to that of tubulin.

The online version of this article includes the following source data and figure supplement(s) for figure 6:

**Source data 1.** PDF file containing original western blots for *Figure 6*, indicating the relevant bands and treatments.

**Source data 2.** Original files for western blot analysis displayed in *Figure 6*.

**Figure supplement 1.** Suppressors influence HIF-1 and FAT-2 levels.

**Figure supplement 1—source data 1.** PDF file containing original western blots for *Figure 6—figure supplement 1*, indicating the relevant bands and treatments.

**Figure supplement 1—source data 2.** Original files for western blot analysis displayed in *Figure 6—figure supplement 1*.

inhibit HIF-1. Finally, the null *ftn-2(et68)* allele caused near-loss (a faint HIF-1 band is occasionally seen) of detectable HIF-1 in *fat-2(wa17)*, suggesting feedback regulation between *ftn-2* and *hif-1* (*Figure 6A–B*, *Figure 6—figure supplement 1A–B*).

Inhibition of *egl-9* promotes HIF-1 activity (*Shao et al., 2009*), which we here verified for *the egl-9(et60)* allele using western blots (*Figure 6A*). Additionally, we found by qPCR that *ftn-2* mRNA levels are as expected reduced by the proposed gain-of-function *hif-1(et69)* allele (*Figure 6C*). We conclude that the *egl-9* and *hif-1* suppressor mutations likely converge on inhibiting *ftn-2* and thus act similarly to the *ftn-2* loss-of-function alleles.

We also used Western blots to evaluate the abundance of the FAT-2 protein expressed from endogenous wild-type or mutant loci, but to which a HA tag was fused using CRISPR/Cas9. We found that the FAT-2::HA levels are severely reduced when the locus contains the S101F substitution present in the *wa17* allele, but are restored close to wild-type levels by the *fat-2(et65)* suppressor mutation (*Figure 6D–E*, *Figure 6—figure supplement 1C–D*). The levels of FAT-2 in the HIF-1 pathway suppressors varied between experiments, with the suppressors sometimes restoring FAT-2 levels and sometimes not, even when the worms were growing well (*Figure 6D–E*, *Figure 6—figure supplement*

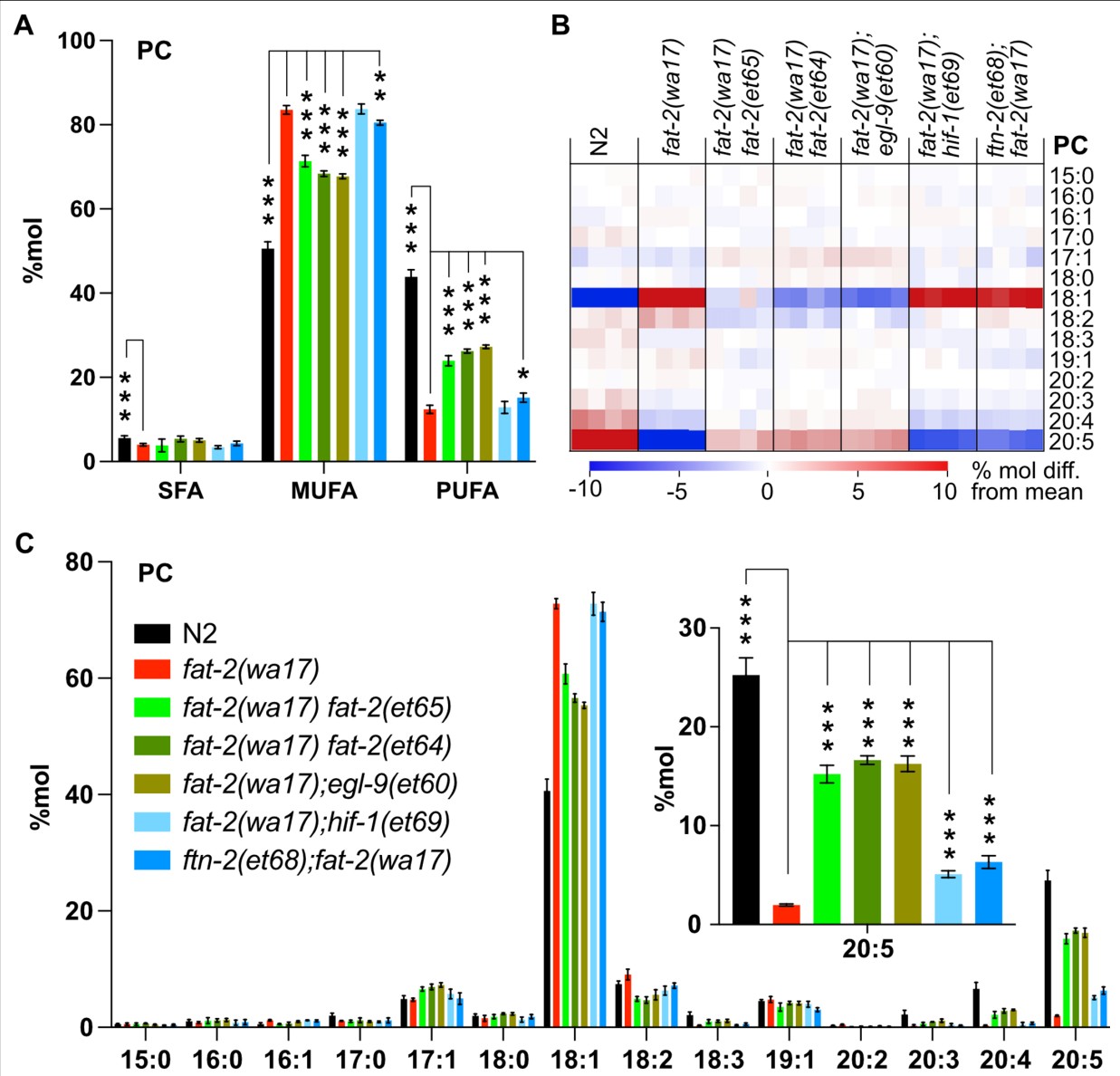

**Figure 7.** Lipidomic analysis of *fat-2(wa17)* suppressors reveals that polyunsaturated fatty acid (PUFA) levels are increased. (**A**) Levels of saturated fatty acids (SFAs), monounsaturated fatty acids (MUFAs), and PUFAs in phosphatidylcholines (PCs) measured in *fat-2(wa17)* suppressors confirm that the suppressors increase PUFA levels in *fat-2(wa17)*. Worms were homozygous for all indicated genotypes, but note that the *hif-1(et69)* allele suppresses *fat-2(wa17)* best in a heterozygous state. (**B**) Heat map analysis of PC species in suppressor mutants. (**C**) Levels of individual FA species in PCs in *fat-2(wa17)* suppressors, the insert shows that levels of C20:5 are significantly increased in all double mutant strains. n=4 populations for each genotype. For A and C (inset), *p<0.05, **p<0.01, ***p<0.001 indicate significant differences compared to the *fat-2(wa17)* control and using one-way ANOVA followed by a Dunnett's multiple comparison test. Note that the N2 and *fat-2(wa17)* samples are the same as in *Figure 3*.

The online version of this article includes the following figure supplement(s) for figure 7:

**Figure supplement 1.** Lipidomics of phosphatidylethanolamines (PEs) in *fat-2(wa17)* suppressors.

*1C–D*). The *fat-2(wa17)* suppressors, except for the intragenic *fat-2* alleles, likely do not act by increasing FAT-2 protein levels.

As already mentioned, ferrous ions ($Fe^{2+}$) are potent activators of desaturases (*Shen et al., 2023*). Given that each of the *fat-2(wa17)* suppressor mutants within the HIF-1 pathway are predicted to ultimately inhibit *ftn-2*, thus increasing the ferrous ion pool, we hypothesized that PUFA levels should be at least partially normalized in the *fat-2(wa17)* suppressors. This was confirmed by lipidomic analysis of phosphatidylcholines (*Figure 7A*) and phosphatidylethanolamines (*Figure 7—figure supplement*

*1A*). In particular, while levels of 18:2 (LA) were not significantly increased in the suppressor strains, the levels of 20:5 (EPA) were significantly increased by more than three folds and to levels near those obtained earlier by supplementing with LA (*Figure 7B–C*, *Figure 7—figure supplement 1B–C*), likely because the suppressor mutations allow *fat-2(wa17)* to produce more LA that is converted by other elongases and desaturases into EPA, the end product.

## Multiple stress response pathways are active in *fat-2(wa17)* and suppressed by *ftn-2(et68)*

The increase in EPA, and PUFA levels in general, likely explains the improved development and growth of *fat-2(wa17)*. We examined other traits that may be rescued by the *fat-2* suppressors, using the *ftn-2(et68)* mutant as a representative because the *egl-9* and *hif-1* alleles converge on it. We found that the membrane fluidity defects in *fat-2(wa17)* were suppressed by *ftn-2(et68)* (*Figure 8A–C*). Additionally, several stress response pathways that are constitutively activated in the *fat-2(wa17)* mutant were also rescued by *ftn-2(et68)*. The mitochondrial UPR visualized with a *hsp-60::GFP* reporter (*Yoneda et al., 2004*) is activated in *fat-2(wa17)* at a level similar to that in *afts-1(et15)*, a known activator of mitochondrial stress (*Rauthan et al., 2013*), and this is suppressed by *ftn-2(et68)* (*Figure 8D–E*). Similarly, the metabolic stress reporter DAF-16::GFP (*Libina et al., 2003*) is constitutively nuclear-localized in *fat-2(wa17)*, and this is also suppressed by *ftn-2(et68)* (*Figure 8F–G*). Using a *hsp-4::GFP* reporter (*Calfon et al., 2002*), we found that the ER UPR is only slightly activated in *fat-2(wa17)* relative to WT (especially in spermatheca), and that this stress response too is partially suppressed by *ftn-2(et68)* (*Figure 8H–I*). Altogether, these results show that the PUFA-deficient *fat-2(wa17)* mutant engages multiple stress response pathways and that these are abated by *ftn-2(et68)*.

## Mimicking *fat-2(wa17)* suppressors using hypoxia or iron supplements

Attempts to mimic the effects of the *fat-2(wa17)* suppressor mutations by hypoxia or supplement treatments were only partially successful. Providing *fat-2(wa17)* with ferric ammonium citrate (FAC), which increases the levels of ferric ions that can be converted into ferrous ions as well as overall iron levels in worms (*Valentini et al., 2012*), provided only a slight rescue of the *fat-2(wa17)* mutant (*Figure 9A*). Additionally, providing *fat-2(wa17)* with ferrous ions in the form of ferrous chloride did not provide any rescue (*Figure 9—figure supplement 1A*). Reducing the levels of ferrous ions with the iron chelator deferoxamine, which we hypothesized would further hinder *fat-2(wa17)* growth, had no effect (*Figure 9—figure supplement 1B*); however, given that the *fat-2(syb7458)* null mutant grows at the same rate as *fat-2(wa17)* in 72 hr (*Figure 9—figure supplement 1C*) but never develops into an adult, we theorize that 72 hr may be too short to see a negative effect from deferoxamine on *fat-2(wa17)*. HIF-1-activating paraquat (PQ; *Hwang et al., 2014*) likewise conferred only a small rescue (*Figure 9B*), while the combination of FAC and PQ did not provide any additional growth rescue, both at 20°C and 25°C (*Figure 9C*; *Figure 9—figure supplement 1D*). The HIF-1 activator hydrogen peroxide (*Xie and Roy, 2012*) also only mildly rescued *fat-2(wa17)* (*Figure 9D*), while two separate hypoxia mimetics (cobalt chloride [*Padmanabha et al., 2015*] and sodium sulfite [*Jiang et al., 2011*]) did not suppress the poor growth of *fat-2(wa17)* (*Figure 9—figure supplement 1E–F*). Additionally, exposing *fat-2(wa17)* to multiple short hypoxia treatments slightly increased growth, but longer hypoxia treatments had no effect (*Figure 9E*). Finally, we also tested a cocktail of eicosanoids, which are derived from PUFAs such as EPA and could be limiting in the *fat-2(wa17)* mutant, but found that they had no rescuing effect when added as a supplement to the culture plates; their half-life and uptake by the worms are unknown (*Figure 9—figure supplement 1G*). Taken together, these results suggest that increasing iron and activating HIF-1 are beneficial to *fat-2(wa17)*, but that achieving physiologically optimal dosing via experimental treatments is difficult.

## Discussion

That dietary PUFAs are essential for mammalian health, with LA and ALA acting as precursors for the synthesis of other PUFAs, is known since the 1930s (*Burr and Burr, 1930*). PUFAs have been linked to several important cellular and physiological processes (reviewed in *Calder, 2012*; *Vrablik and Watts, 2013*; *Harayama and Shimizu, 2020*), including cell membrane properties and organelle dynamics (*Antonny et al., 2015*), autophagy (*O'Rourke et al., 2013*), mitochondria function (*Stanley et al.,*

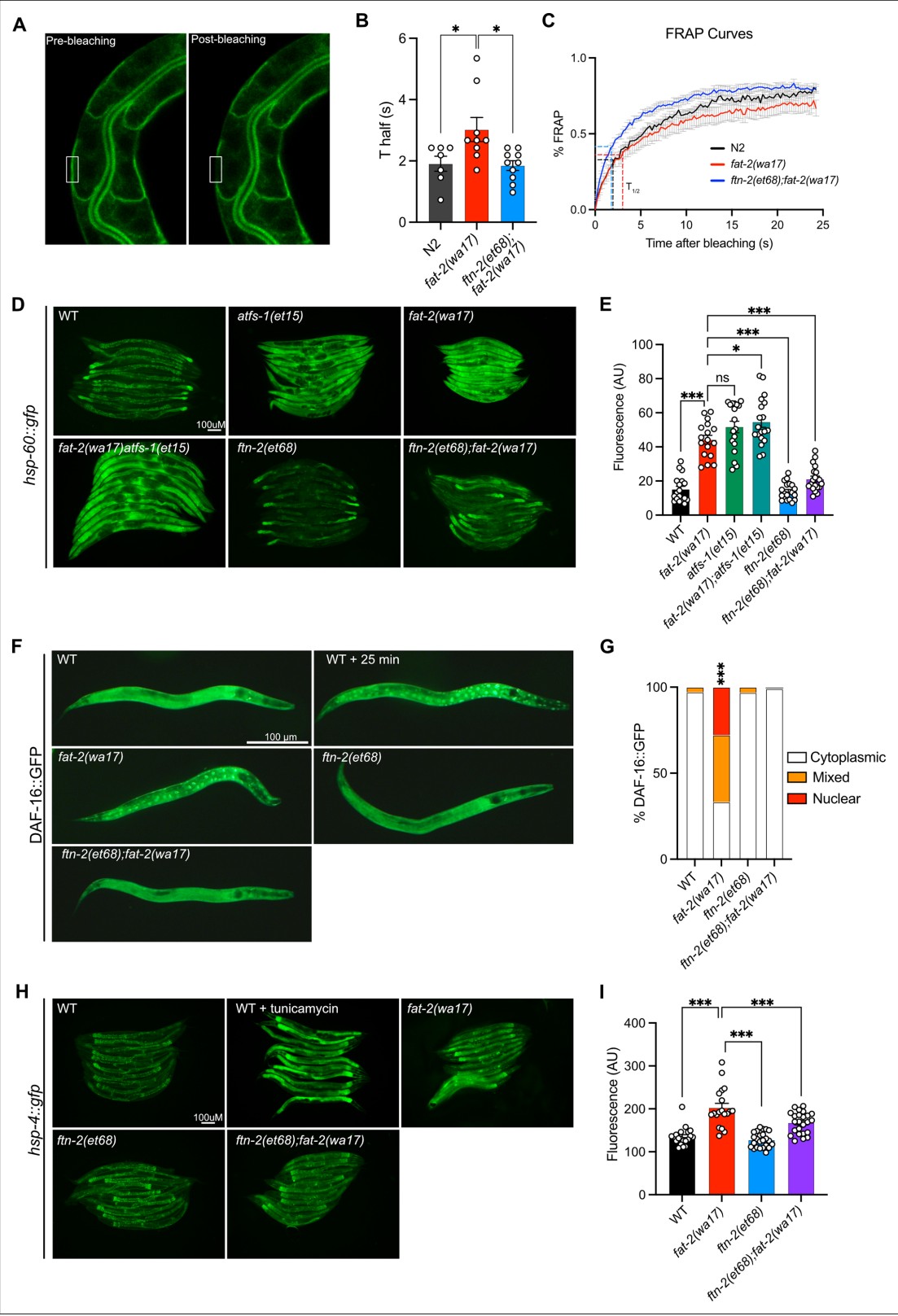

**Figure 8.** *ftn-2(et68)* rescues *fat-2(wa17)*'s stress responses. (**A**) Representative image of a fluorescence recovery after photobleaching (FRAP) experiment, showing pGLO-1::GFP-CAAX-positive intestinal membranes. The rectangle indicates the bleached area. (**B–C**) T$_{half}$ values and FRAP curves show that *ftn-2(et68);fat-2(wa17)* has less rigid membranes than *fat-2(wa17)*. From left to right, n=7, 9, 10. (**D–E**) Representative images and quantification of *ftn-2(et68)* rescue of *fat-2(wa17)* mitochondrial stress with a *hsp-60::gfp* reporter. *atfs-1(et15)* serves as a control for high mitochondrial UPR activation.

*Figure 8 continued on next page*

*Figure 8 continued*

n=20 for each genotype. (**F–G**) Representative images and quantification of DAF-16::GFP localization showing that the DAF-16 stress response is constitutively active in the *fat-2(wa17)* mutant but normalized by *ftn-2(et68)*. Chi-squared test shows that *fat-2(wa17)* is significantly different from wild-type (WT). n=100 for each genotype. (**H–I**) Representative images and quantification of mild ER stress in *fat-2(wa17)* that is slightly rescued by *ftn-2(et68)* using a *hsp-4::gfp* reporter. n=20 for each genotype. *p<0.05, **p<0.01, ***p<0.001 indicate significant differences compared to the *fat-2(wa17)* control (ordinary one-way ANOVA with Tukey multiple comparisons test).

*2012*), ferroptosis (*Yang et al., 2016*; *Lee et al., 2020*; *Perez et al., 2020*), regulation of the *daf-2/*insulin, mTOR and p38-MAPK pathways (*Horikawa and Sakamoto, 2010*; *Chamoli et al., 2020*; *Liu et al., 2020*), SREBP stability and signaling (*Worgall et al., 1998*; *Yahagi et al., 1999*), lipid droplet fusion (*Wang et al., 2022*), neuronal signaling and neurotransmission (*Kahn-Kirby et al., 2004*; *Marza and Lesa, 2006*; *Vásquez et al., 2014*), TRPV-dependent sensory signaling (*Kahn-Kirby et al., 2004*), oocyte development (*Chen et al., 2016*), and telomere length (*Wu et al., 2024*). Which of these, if any, is the specific essential role of PUFAs in animal physiology? And are there molecular mechanisms that can compensate for PUFA deficiency? In the present study, we approached these questions using

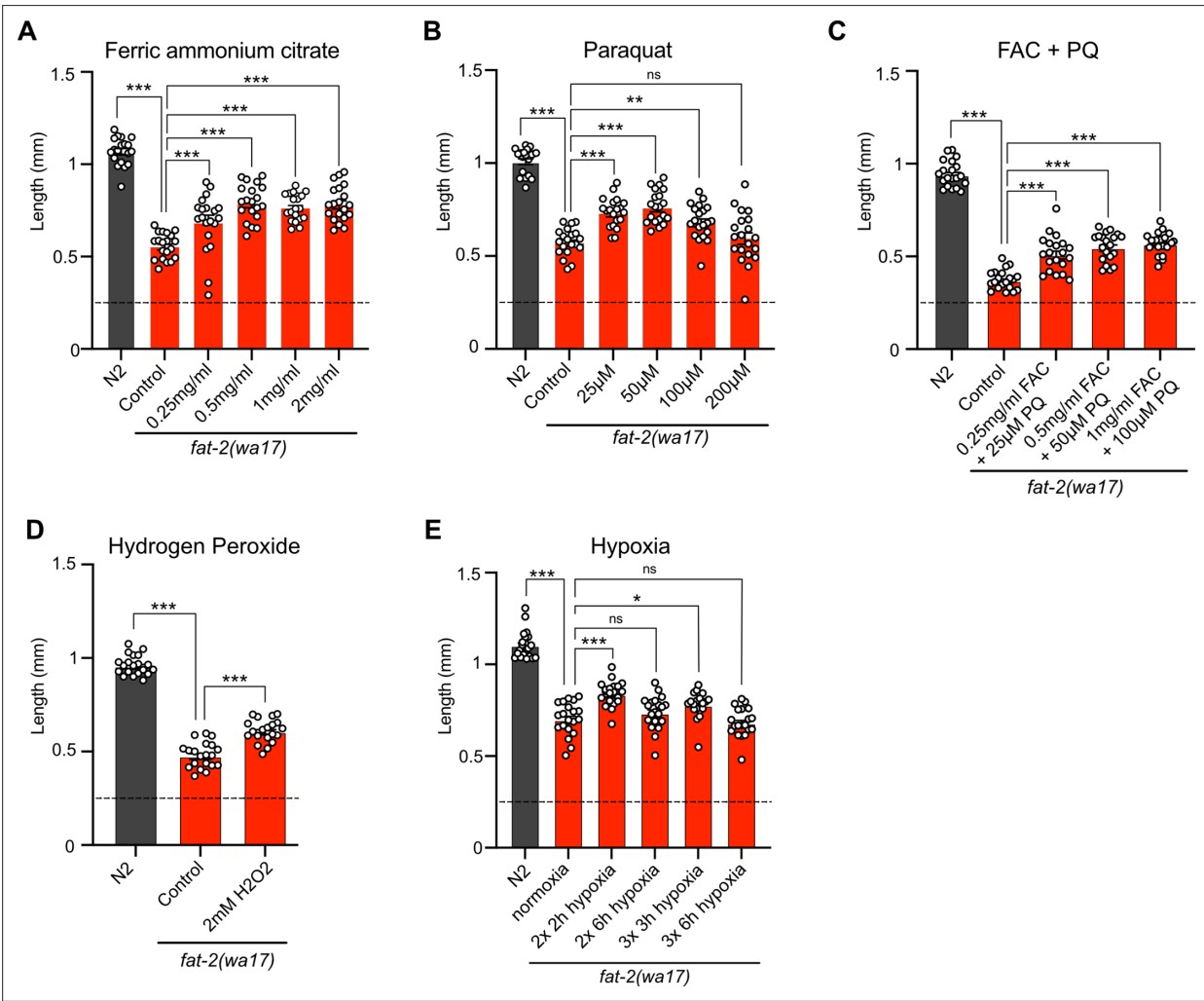

**Figure 9.** Exogenous treatments that mimic *fat-2(wa17)* suppressors partially rescue *fat-2(wa17)*. (**A–E**) Length assay of *fat-2(wa17)* cultivated with different treatments for 72 hr after L1 stage synchronization. The horizontal dashed line represents the approximate length of L1 worms at the start of each experiment. n=20 for each genotype/condition. *p<0.05, **p<0.01, ***p<0.001 indicate significant differences compared to the *fat-2(wa17)* control (ordinary one-way ANOVA with Tukey multiple comparisons test).

The online version of this article includes the following figure supplement(s) for figure 9:

**Figure supplement 1.** Exogenous treatment of *fat-2(wa17)* to mimic suppressors.

forward genetics in *C. elegans*. While *C. elegans* can de novo synthesize PUFAs, mutations that impair the production of certain PUFAs can lead to developmental defects or lethality (*Watts and Browse, 2002*; *Dancy et al., 2015*; *Perez and Van Gilst, 2008*), offering opportunities for suppressor screens. Here, we showed that defects in the *fat-2(wa17)* mutant, which has limited Δ12 desaturase activity and only produces trace amounts of PUFAs, are suppressed by either compensatory intragenic mutations within *fat-2* itself or by mutations within the HIF-1 pathway. The fact that screening approximately 40,000 haploid genomes for *fat-2(wa17)* suppressors and finding mutations only within *fat-2* itself or within the HIF-1 pathway suggests that the screen has reached near-saturation and that we may have identified most, if not all, possible genetic ways to compensate for the *fat-2(wa17)* mutation. Importantly, none of the *fat-2(wa17)* suppressor mutations that we identified compensate for the PUFA shortage itself. Instead, the *fat-2(wa17)* suppressors act by boosting desaturase activity to allow the *fat-2(wa17)* mutant to synthesize more PUFAs; the *fat-2(wa17)* suppressors, therefore, cannot suppress the defects of the *fat-2* null mutant, as we specifically showed for *ftn-2(et68)*. We draw the important conclusion that PUFAs are not only essential but also that their essential functions cannot be genetically replaced.

The *fat-2(wa17)* suppressor mutations within the HIF-1 pathway converge on the inhibition of *ftn-2*. The primary function of ferritin is to provide a harmless storage of iron within cells: ferritin promotes the oxidation of ferrous ions and stores the resulting ferric ions in a mineralized form (*Plays et al., 2021*). Thus, *ftn-2* inhibition results in reduced total cellular iron but increased levels of ferrous ions, i.e., $Fe^{2+}$ (*Jenkins et al., 2020*; *Pekec et al., 2022*). Importantly, ferrous ions are required for desaturase reactions, and increasing ferrous ion concentration is a potent way to increase activity because it accelerates the rate at which the desaturase cycles from the inactive post-reaction $Fe^{3+}$-bound state to the active $Fe^{2+}$-bound state (*Shen et al., 2023*; *Shen et al., 2020*). Because eukaryotic desaturases are all evolutionarily closely related (*Sperling et al., 2003*) and act in essentially the same way, ferrous ions must also be potent FAT-2 activators and thus boost the output from the mutant FAT-2(S101F) protein produced by the *fat-2(wa17)* allele or from the reduced FAT-2 protein levels in *fat-2* RNAi-treated worms. Our findings suggest an elegant explanation for the observation that HIF-1 inhibits *ftn-2* expression in *C. elegans* (*Romney et al., 2011*): this is likely an adaptive response to boost desaturase activity when oxygen or iron is limiting, ensuring a maximum output under adverse conditions. $Fe^{2+}$ and HIF may contribute to desaturase boost also in human since CytB5 (which supplies $Fe^{2+}$ to desaturases) promotes SFA tolerance while VHL (which causes HIF degradation) prevents SFA tolerance in cultured cells (*Zhu et al., 2019*). Other mechanisms are also possible. For example, mutations in the HIF-1 pathway could somehow reduce EPA turnover rates in the *fat-2(wa17)* mutant and allow its levels to rise above an essential threshold. This hypothesis is consistent with the observation that the suppressors can rescue both the *fat-2(wa17)* mutant and *fat-2* RNAi-treated worms but not the *fat-2* null mutant. It is even possible, though deemed unlikely, that the *fat-2(wa17)* suppressors act by compensating for the PUFA shortage via some undefined separate process downstream of the lipid changes and that they only indirectly result in elevated EPA levels.

Lipidomic analysis showed that among all PUFAs, it was the EPA levels that were best restored by the *fat-2(wa17)* suppressors. It is likely that any LA molecule produced in the mutants is quickly acted upon by downstream desaturases and elongases, leading to increased levels of the end product, namely EPA. EPA may be a sufficient or particularly important PUFA for sustaining *C. elegans* health, given that the *fat-2(wa17)* mutant is well rescued by EPA supplements. Indeed, DHA, which is not produced by *C. elegans*, is also able to rescue the *fat-2(wa17)* mutant. Others have shown that supplementing nearly completely EPA-deficient *fat-3 C. elegans* mutants with DHA significantly restored their EPA levels, suggesting that DHA supplements reduce EPA turnover (*Lesa et al., 2003*). EPA and DHA, being long chain PUFAs should have similar fluidizing effects on membrane properties (though in vitro experiments challenge this view; *Sherratt et al., 2021*), and both can serve as precursors of eicosanoids or docosanoids, particularly inflammatory ones (*Chiang and Serhan, 2020*). Abundant literature indicates that EPA is a particularly important PUFA in *C. elegans*. Phosphatidylcholines containing two attached EPA molecules are very abundant in *C. elegans* membranes, and their abundance increases the most in response to a temperature shift from 25–15°C, suggesting an important role in fluidity homeostasis (*Tanaka et al., 1999*). Long chain PUFAs such as EPA are required for efficient neurotransmission in *C. elegans*: mutants unable to produce them have depleted levels of synaptic vesicles accompanied by poor motility, and these defects are rescued by exogenous PUFAs,

including DHA (*Lesa et al., 2003*). *C. elegans* can also convert EPA to eicosanoids in a cytochrome P450-dependent manner (*Kulas et al., 2008*; *Mokoena et al., 2020*); inhibiting this process results in reduced pharyngeal pumping rate, suggesting that regulation of muscular contraction by eicosanoids is conserved from nematodes to mammals (*Kosel et al., 2011*). EPA-derived eicosanoids are also required for guiding some cell migrations in *C. elegans*, including that of sperm (*Hoang et al., 2013*). EPA, and other PUFAs, can inhibit the nuclear localization of DAF-16 in *fat-2*-RNAi-treated worms, suggesting that they mediate signaling via this insulin receptor homolog and thus generally promote growth in *C. elegans* rather than stress resistance and fat storage (*Horikawa and Sakamoto, 2010*). In conclusion, EPA is clearly an important PUFA in *C. elegans,* and our work suggests that its multifaceted functions cannot be replaced by mutations in any one gene.

Finally, the case of the three novel *egl-9* alleles isolated in our screen deserves special attention. All three alleles specifically affect the arginine at position 557 of the EGL-9 protein (it is converted to a histidine in two of the alleles, and to a cysteine in the third). This Arg557 in EGL-9 is specifically required for its ability to hydroxylate HIF-1, thus marking it for ubiquitin-dependent degradation (*Epstein et al., 2001*). Null alleles of *egl-9* were not picked in our screen, and directly testing such a null allele revealed it to be ineffective as a *fat-2(wa17)* suppressor. We conclude that the EGL-9 proteins bearing a mutation at position Arg557 retain important functions while being unable to hydroxylate HIF-1. Others have previously demonstrated that EGL-9 could inhibit HIF-1 even when unable to hydroxylate it (*Shao et al., 2009*). Clearly, there is more to EGL-9 than its function as a HIF-1 hydroxylase, and it would be interesting in the future to detail this further.

We conclude that PUFA-deficient *fat-2(wa17)* mutants benefit only slightly from membrane-fluidizing treatments, and that there is likely no genetic way to compensate for PUFA deficiency. *fat-2(wa17)* mutants can only be rescued by boosting the activity of its defective desaturase, and restoring EPA levels are likely sufficient to suppress most *fat-2(wa17)* phenotypes suggesting a particularly important role for this PUFA in *C. elegans*. In the future, it will be interesting to determine if boosting desaturase activity by inhibition of ferritin expression via HIF-1 is also a beneficial response to hypoxia in worms and humans.

## Materials and methods

### *C. elegans* strains and cultivation

The wild-type *C. elegans* reference strain N2, *fat-2(wa17)*, *nhr-49(et8)*, *mdt-15(et14)*, *paqr-1(et52)*, *paqr-2(3410)*, *acs-13(et54)*, *fld-1(et46)*, *hif-1(ok2564)*, *ftn-2(ok404)*, *ftn-1(ok3625)*, *egl-9(sa307)*, *atfs-1(et15)*, *zcIs4 [hsp-4::GFP]*, *zcIs9 [hsp-60::GFP +lin-15(+)]* and *zIs356 [daf-16p::daf-16a/b::GFP +rol-6(su1006)]* are available from the *C. elegans* Genetics Center (CGC; USA). The PHX7548 (*fat-2(syb7458)/nT1[qIs51](IV;V)*) strain was created by Suny Biotech Co using CRISPR/Cas9 and carries a deletion of 1387 bp between flanking sequences 5'-aaacttggcccccgacgaagatg-3' and 5'-gtgataatgacgagaataagtcct-3'. *fat-2(syb7458)* worms were maintained in an unbalanced state on non-peptone plates containing OP50 grown overnight in LB containing 2 mM linoleic acid.

Unless otherwise stated, experiments were performed at 20 °C, using the *E. coli* strain OP50 as a food source, which was re-streaked every 6–8 wk and maintained on LB plates at 4°C. Single colonies were cultivated overnight at 37 °C in LB medium before being used to seed NGM plates. Stock solutions of supplements were filter-sterilized and added to cooled NGM after autoclaving to produce supplement plates.

### Construction of *fat-2(+)*

The *pfat-2(+)* construct was generated with the NEB PCR Cloning Kit for amplification of *fat-2(+)* with the following primers: 5'-gagctcaagaagcgtttcca-3' and 5'-gggcaagaatttgtagtgtca-3' using N2 genomic DNA. Plasmids were prepared with a GeneJet Plasmid Miniprep Kit and injected at the following concentrations: *pfat-2(+)* of 20 µg/µl, *pRF4(rol-6)* of 40 µg/µl, and *pBSKS* of 35 µg/µl into *fat-2(wa17)* and N2 worms.

### Pre-loading of *E. coli* with fatty acids or eicosanoids

Stock solutions of fatty acids (Merck) or eicosanoids (Primary Eicosanoid HPLC Mixture; Cayman Chemical) dissolved in ethanol (EPA, DHA, OA), DMSO (LA), or methyl acetate (eicosanoids) were

diluted in LB media to the appropriate final concentration, inoculated with OP50 bacteria, and shaken overnight at 37 °C. The bacteria were washed twice in M9 to remove fatty acids, concentrated 10 X by centrifugation, dissolved in LB, and seeded onto NGM plates lacking peptone.

## Growth assays

For length measurement studies, synchronized L1s were plated onto test plates seeded with OP50, and worms were mounted and photographed 72 hr later. Experiments performed at 15 °C were photographed after 144 hr. The length of 20 worms was measured using ImageJ.

For hydrogen peroxide treatment, synchronized worms were incubated in 2 mM hydrogen peroxide for 2 hr at the L1 stage before being plated on NGM plates for 72 hr. For hypoxia treatment, synchronized L1s were incubated for 2–6 hr in a hypoxia chamber, returned to normoxia for 24 hr, and hypoxia exposure was repeated as stated in the figure.

## Oil Red O staining

Synchronized day 1 adults were washed three times with PBST and fixed for 3 min in 60% isopropanol. Worms were then rotated for 2 hr in filtered 60% Oil Red O staining solution. The stained worms were washed three times in PBST before being mounted on agarose pads and imaged with a Zeiss Axioscope microscope.

## Mutagenesis and screen for *fat-2(wa17)* suppressors

*fat-2(wa17)* worms were mutagenized for 4 hr by incubation in the presence of 0.05 M ethyl methane sulfonate (EMS) according to the standard protocol (*Sulston and Hodgkin, 1988*). The worms were then washed and spotted onto an NGM plate. After 2 hr, L4 hermaphrodite animals were transferred to new plates. 8–10 d later, F1 progeny were bleached, washed, and their eggs allowed to hatch overnight in M9. The resulting L1 larvae were spotted onto new plates, cultivated at 20 °C, then screened after 72 hr for gravid F2 worms, which were then picked to new plates for further analysis. In total, approximately 40, 000 independently mutagenized haploid genomes were screened. The isolated suppressor mutants were outcrossed 4–6 times prior to whole genome sequencing, and 8–10 times prior to characterization. Outcrossing was done by mating N2 males to a suppressor, then crossing male progeny to *fat-2(wa17)* mutant worms. Progeny from this cross were picked to individual plates and kept at 20 °C, then screened for *fat-2(wa17)* homozygosity using PCR, followed by testing the F2 progeny for ability to grow to adults in 72 hr. Genotyping primers for the suppressor mutants are included in *Supplementary file 1*.

## Whole genome sequencing

The genomes of the ten suppressor mutants that had been outcrossed 4 or 6 times were sequenced by Eurofins (Constance, Germany) with a mean coverage varying from 40.68X to 63.05X and their genomes assembled using the *C. elegans* genome version cel235 from Ensembl (REF: PMID: 37953337). Eurofins applied customised filters to the variants to filter false positives using GATK's Variant Filtration module (*DePristo et al., 2011*; *McKenna et al., 2010*). Variants detected were annotated based on their gene context using snpEff (*Cingolani et al., 2012*). For each suppressor mutant, one or two hot spots, i.e., small genomic area containing several mutations, were identified and candidate mutations tested experimentally as described in the text.

## CRISPR-Cas9 genome editing

The recreation of the candidate suppressor mutations and insertion of the 3xFLAG tag into the *hif-1* gene was performed using CRISPR-Cas9 gene editing as previously described (*Ghanta and Mello, 2020*; *Dokshin et al., 2018*). The insertion of the ssDNA oligos was performed utilizing the homology-directed repair (HDR) mechanisms. The protospacer-adjacent motif (PAM) site of the ssDNA oligo template was flanked by 40 bp homology arms. Design and synthesis of the ssDNA and CRISPR RNA (crRNA) was performed using the Alt-R HDR Design Tool from IDT (Integrated DNA Technologies, Inc; Coralville, IA, USA), including proprietary modifications that improve oligo stability. To recreate the *hif-1(et69)* allele, we used the crRNA sequence 5'- UUUCUUAACGUGUGUAUUUCGUUUUAGA GCUAUGCU-3' and the DNA oligo donor sequence 5'-AGTTCCATACATTTAGCAAGTGATTTCTTAAC GTGTGTATTTCAAGAGCACGTAAGAACAGCTACGATGACGTTTTGCAATGGCT-3. To introduce the

3xFLAG at the N-terminus coding end of *hif-1,* we used the crRNA sequence 5'- GAAAAUAAUCAA GAGAGCAUGUUUUAGAGCUAUGCU-3' and the DNA oligo donor sequence 5'-AAATGAACAACA GCCTAGTTCTTATTCCCCATTTCCAATGCTCTCTGACTACAAGGACCACGACGGCGATTATAAG GATCACGACATCGACTACAAAGACGACGATGACAAGTGATTATTTTCTACCCCCTCTCAAACTG TTCATTGTTTTG-3'. The injection mixes were prepared using 10 µg/µl of the Cas9 enzyme (IDT), 0.4 µg/µl tracrRNA (IDT), 2.8 0.4 µg/µl crRNA (IDT), 1 µg/µl of ssDNA (IDT), and 40 ng/µl of PRF4(*rol-6*) or Pmyo-2(GFP) plasmid. The mixture was microinjected into the posterior gonad of the worm and the F1 generation was screened for animals expressing the reporter plasmid. Genotypes were tested by PCR and successfully edited genes were confirmed by Sanger sequencing (Eurofins).

## Fluorescence recovery after photobleaching (FRAP)

FRAP experiments were carried out using a membrane-associated prenylated GFP reporter expressed in intestinal cells as previously described (*Devkota and Pilon, 2018*), using a Zeiss LSM700inv laser scanning confocal microscope with a 40 X water immersion objective. Briefly, the GFP-positive membranes were photobleached over a rectangular area (30×4 pixels) using 30 iterations of the 488 nm laser with 50% laser power transmission. Images were collected at a 12-bit intensity resolution over 256×256 pixels (digital zoom 4 X) using a pixel dwell time of 1.58 µs, and were all acquired under identical settings. The recovery of fluorescence was traced for 25 s. Fluorescence recovery and $T_{half}$ were calculated as previously described (*Svensk et al., 2016*).

## Stress response assay

Worms were imaged with a Zeiss Axioscope and fluorescence intensity was quantified with ImageJ (n≥20 for all experiments). Worm strains carrying *hsp-60::GFP* were imaged as day 1 adults, and the fluorescence values were taken from a 39 µm circumference circle in the brightest part of the anterior part of the worm. Worm strains carrying *DAF-16::GFP* were imaged as L4s and the percentage of worms with cytoplasmic or nuclear localization was quantified. Worm strains carrying *hsp-4::GFP* were imaged as day 1 adults and the fluorescence of the whole worm was quantified.

## Lipidomics

Samples were composed of synchronized L4 larvae (one 9 cm diameter plate/sample; each treatment/ genotype was prepared in four independently grown replicates) grown on NGM or non-peptone plates seeded with linoleic acid. In the case of LA to NGM samples, worms were grown until late L3/ early L4 stage on linoleic acid seeded non-peptone plates before being transferred to NGM plates for 6 hr before collection. Worms were washed three times in M9, pelleted and stored at –80 °C until analysis. For lipid extraction, the pellet was sonicated for 10 min in methanol;butanol [1:3] and then extracted according to published methods (*Löfgren et al., 2016*). Lipid extracts were evaporated and reconstituted in chloroform:methanol [1:2] with 5 mM ammonium acetate. This solution was infused directly (shotgun approach) into a QTRAP 5500 mass spectrometer (Sciex) equipped with a TriVersa NanoMate (Advion Bioscience) as described previously (*Jung et al., 2011*). Phospholipids were measured using precursor ion scanning in negative mode using the fatty acids as fragments (*Ekroos et al., 2003*; *Ejsing et al., 2009*). To generate the phospholipid composition (as mol%), the signals from individual phospholipids (area under the m/z peak in the spectra) were divided by the signal from all detected phospholipids of the same class. The data were evaluated using the LipidView software (Sciex). The data were further analyzed using Qlucore Omics Explorer n.n (Qlucore AB) for analysis. The data were normalized for the purpose of the heat map visualization (mean = 0; variance = 1).

## Protein extraction and western blots

Worms were lysed using lysis buffer containing 25 mM Tris (pH 7.5), 300 mM NaCl, 0.1% NP40, and 1 X protease inhibitor on ice with a motorized pestle. Samples were centrifuged at 20,000 g for 15 min at 4°C, and protein sample concentration was quantified using the BCA protein assay kit. 15 µg of protein were mixed with Laemmli sample loading buffer containing β-mercaptoethanol, boiled for 10 min, and loaded in 4–20% gradient precast SDS gel. After electrophoresis, the proteins were transferred to nitrocellulose membranes using Trans-Blot Turbo Transfer Packs and a Trans-Blot Turbo apparatus/predefined mixed-MW program. Blots were blocked in 5% nonfat dry milk in PBST for 1 hr at room temperature. Blots were incubated with primary antibodies overnight at 4 °C (mouse

monoclonal anti-FLAG antibody (M2, Sigma Aldrich) 1:5000 dilution) or 1 hr at room temperature (mouse monoclonal anti-alpha-Tubulin (B512, Sigma Aldrich) 1:5000 dilution). Blots were then washed with PBST and incubated with swine-anti-rabbit HRP 1:3000 dilution or goat anti-mouse HRP 1:3000 dilution for 1 hr at room temperature and washed again with PBST. Detection of the hybridized antibody was performed using an ECL detection kit (Immobilon Western, Millipore), and the signal was visualized with a digital camera (VersaDoc, Bio-Rad).

### Quantitative PCR (qPCR)

Total cellular worm RNA was isolated using the RNeasy Plus Kit according to the manufacturer's instructions (Qiagen) and quantified using a NanoDrop spectrophotometer (ND-1000; Thermo Scientific). cDNA was obtained using the RevertAid H Minus First Strand cDNA Synthesis Kit (Thermo Scientific) with random hexamers. qPCR was performed with a CFX Connect thermal cycler (Bio-Rad) using HOT FIREpol EvaGreen qPCR SuperMix (Solis Biodyne) and standard primers. Samples were measured in triplicates. The relative expression of each gene was calculated according to the delta-delta CT method. Expression of the housekeeping gene *tba-1* was used to normalize for variations in RNA input. Primers used were as follows: *ftn-2*, forward (TACCACTCCGAGGTTGAAGC) and reverse (TGGAAGGGCAACATCGTCAC); *tba-1*, forward (TCTCGCAGGTTGTGTCTTCC) and reverse (AGCCTCATGGTAAGCCTGAA).

### Statistics

Error bars for the worm length measurements show the standard error of the mean, and one-way ANOVA tests were used to identify significant differences from *fat-2(wa17)* control, unless otherwise stated. All experiments were independently repeated at least twice with similar results, and the statistics shown apply to the presented experimental results.

### Materials availability statement

Strains generated in this study will be deposited at the *Caenorhabditis* Genetics Center (CGC). Strains, and other reagents, can also be requested from the corresponding author.

## Acknowledgements

Some strains were provided by the CGC, which is funded by the NIH Office of Research Infrastructure Programs (P40 OD010440). This research was funded by Cancerfonden (22 1984 Pj) and Vetenskapsrådet (2024–04012).

## Additional information

### Funding

| Funder | Grant reference number | Author |
|---|---|---|
| Cancerfonden | 22 1984 Pj | Marc Pilon |
| Vetenskapsrådet | 2024-04012 | Marc Pilon |
| NIH Office of Research Infrastructure Programs | P40 OD010440 | |

The funders had no role in study design, data collection and interpretation, or the decision to submit the work for publication.

### Author contributions

Delaney Kaper, Conceptualization, Data curation, Formal analysis, Investigation, Visualization, Writing – original draft, Writing – review and editing; Uroš Radović, Conceptualization, Data curation, Formal analysis, Investigation, Visualization; Per-Olof Bergh, Marcus Henricsson, Investigation; August Qvist, Formal analysis, Investigation, Visualization; Jan Borén, Project administration; Marc Pilon, Conceptualization, Data curation, Formal analysis, Funding acquisition, Investigation, Visualization, Writing – original draft, Project administration, Writing – review and editing

## Author ORCIDs
Delaney Kaper ⓘ https://orcid.org/0000-0002-9929-8104
Uroš Radović ⓘ https://orcid.org/0009-0002-9008-0202
August Qvist ⓘ https://orcid.org/0009-0005-9566-9026
Marc Pilon ⓘ https://orcid.org/0000-0003-3919-2882

Reviewer #1 (Public review): https://doi.org/10.7554/eLife.104181.4.sa1
Reviewer #2 (Public review): https://doi.org/10.7554/eLife.104181.4.sa2
Author response https://doi.org/10.7554/eLife.104181.4.sa3

## Additional files

### Supplementary files
Supplementary file 1. Primers for genotyping.

MDAR checklist

### Data availability
The targetted lipidomics data for Figures 3 and 7 (and their associated figure supplements) is provided as Source Data.

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
