## [Editor Report · eLife Assessment]

This **fundamental** study investigates the role of polyunsaturated fatty acids (PUFAs) in physiology and membrane biology, using a unique model to perform a thorough genetic screen that demonstrates that PUFA synthesis defects cannot be compensated for by mutations in other pathways. These findings are supported by **compelling** evidence from a high quality genetic screen, functional validation of their hits, and lipid analyses. This study will appeal to researchers in membrane biology, lipid metabolism, and *C. elegans* genetics.

---

## [Referee Report · Reviewer #1 (Public review)]

Summary:

This study addresses the roles of polyunsaturated fatty acids (PUFAs) in animal physiology and membrane function. A *C. elegans* strain carrying the fat-2(wa17) mutation possesses a very limited ability to synthesize PUFAs and there is no dietary input because the *E. coli* diet consumed by lab grown *C. elegans* does not contain any PUFAs. The fat-2 mutant strain was characterized to confirm that the worms grow slowly, have rigid membranes, and have a constitutive mitochondrial stress response. The authors showed that chemical treatments or mutations known to increase membrane fluidity did not rescue growth defects. A thorough genetic screen was performed to identify genetic changes to compensate for the lack of PUFAs. The newly isolated suppressor mutations that compensated for FAT-2 growth defects included intergenic suppressors in the fat-2 gene, as well as constitutive mutations in the hypoxia sensing pathway components EGL-9 and HIF-1, and loss of function mutations in ftn-2, a gene encoding the iron storage protein ferritin. Taken together, these mutations lead to the model that increased intracellular iron, an essential cofactor for fatty acid desaturases, allows the minimally functional FAT-2(wa17) enzyme to be more active, resulting in increased desaturation and increased PUFA synthesis.

Strengths:

(1) This study provides new information further characterizing fat-2 mutants. The authors measured increased rigidity of membranes compared to wild type worms, however this rigidity is not able to be rescued with other fluidity treatments such as detergent or mutants. Rescue was only achieved with polyunsaturated fatty acid supplementation.

(2) A very thorough genetic suppressor screen was performed. In addition to some internal fat-2 compensatory mutations, the only changes in pathways identified that are capable of compensating for deficient PUFA synthesis was the hypoxia pathway and the iron storage protein ferritin. Suppressor mutations included an egl-9 mutation that constitutively activates HIF-1, and Gain of function mutations in hif-1 that are dominant. This increased activity of HIF conferred by specific egl-9 and hif-1 mutations lead to decreased expression of ftn-2. Indeed, loss of ftn-2 leads to higher intracellular iron. The increased iron apparently makes the FAT-2 fatty acid desaturase enzyme more active, allowing for the production of more PUFAs.

(3) The mutations isolated in the suppressor screen show that the only mutations able to compensate for lack of PUFAs were ones that increased PUFA synthesis by the defective FAT-2 desaturase, thus demonstrating the essential need for PUFAs that cannot be overcome by changes in other pathways. This is a very novel study, taking advantage of genetic analysis of *C. elegans*, and it confirms the observations in humans that certain essential PUFAs are required for growth and development.

(4) Overall, the paper is well written, and the experiments were carried out carefully and thoroughly. The conclusions are well supported by the results.

Weaknesses:

Overall, there are not many weaknesses. The main one I noticed is that the lipidomic analysis shown in Figs 3C, 7C, S1 and S3. While these data are an essential part of the analysis and provide strong evidence for the conclusions of the study, it is unfortunate that the methods used did not enable the distinction between two 18:1 isomers. These two isomers of 18:1 are important in *C. elegans* biology, because one is a substrate for FAT-2 (18:1n-9, oleic acid) and the other is not (18:1n-7, cis vaccenic acid). Although rarer in mammals, cis-vaccenic acid is the most abundant fatty acid in *C. elegans* and is likely the most important structural MUFA. The measurement of these two isomers is not essential for the conclusions of the study.

---

## [Referee Report · Reviewer #2 (Public review)]

Summary:

The authors use a genetic screen in *C. elegans* to investigate the physiological roles of polyunsaturated fatty acids (PUFAs). They screen for mutations that rescue fat-2 mutants, which have strong reductions in PUFAs. As a result, either mutations in fat-2 itself or mutations in genes involved in the HIF-1 pathway were found to rescue fat-2 mutants. Mutants in the HIF-1 pathway rescue fat-2 mutants by boosting their catalytic activity (via upregulated Fe2+). Thus, the authors show that in the context of fat-2 mutation, the sole genetic means to rescue PUFA insufficiency is to restore PUFA levels.

Strengths:

As *C. elegans* can produce PUFAs de novo as essential lipids, the genetic model is well-suited to study the fundamental roles of PUFAs. The genetic screen finds mutations in convergent pathways, suggesting that it has reached near-saturation. The authors extensively validate the results of the screening and provide sufficient mechanistic insights to show how PUFA levels are restored in HIF-1 pathway mutants. As many of the mutations found to rescue fat-2 mutants are of gain-of-function, it is unlikely that similar discoveries could have been made with other approaches like genome-wide CRISPR screenings, making the current study distinctive. Consequently, the study provides important messages. First, it shows that PUFAs are essential for life. The inability to genetically rescue PUFA deficiency, except for mutations that restore PUFA levels, suggests that they have pleiotropic essential functions. In addition, the results suggest that the most essential functions of PUFAs are not in fluidity regulation, which is consistent with recent reviews proposing that the importance of unsaturation goes beyond fluidity (doi: 10.1016/j.tibs.2023.08.004 and doi: 10.1101/cshperspect.a041409). Thus, the study provides fundamental insights about how membrane lipid composition can be linked to biological functions.

Weaknesses:

The authors put in a lot of effort to answer the questions that arose through peer review, and now all the claims seem to be supported by experimental data. Thus, I do not see obvious weaknesses. Of course, it remains unclear what PUFAs do beyond fluidity regulation, but this is something that cannot be answered from a single study.

---

## [Author Response]

The following is the authors’ response to the previous reviews.

**Reviewer #1:**
The addition of the discussion about the two isomers of 18:1 didn't quite work in the place that the authors added. What the authors wrote on line 126 is true about 18:1 isomers in wild type worms. However, they are reporting their lipidomics results of the fat-2(wa17) mutant worms. In this case, a substantial amount of the 18:1 is the oleic acid (18:1n-9) isomer. The authors can check Table 2 in their reference [10] and see that wild type and other fat mutants indeed contain approximately 10 fold more cis vaccenic than oleic acid, the fat-2(wa17) mutants do accumulate oleic acid, because the wild type activity of FAT-2 is to convert oleic acid to linoleic acid, where it can be converted to downstream PUFAs. I suggest editing their sentence on line 126 to say that the high 18:1 they observed agrees with [10], and then comment about reference 10 showing the majority of 18:1 being the cis-vaccenic isomer in most strains, but the oleic acid isomer is more abundantly in the fat-2(wa17) mutant strain.

We thank the reviewer for spotting that and sparing us a bit of embarrassment. We have now modified the text and hope we got it right this time:

"Even though the lipid analysis methods used here are not able to distinguish between different 18:1 species, a previous study showed that the majority of the 18:1 fatty acids in the *fat-2(wa17)* mutant is actually 18:1n9 (OA) [10] and not 18:1n7 (vaccenic acid) as in most other strains [10,23]; this is because OA is the substrate of FAT-2 and thus accumulates in the mutant."

**Reviewer #2:**
I still do not agree with the answer to my previous comment 6 regarding Figure S2E. The authors claim that hif-1(et69) suppresses fat-2(wa17) in a ftn-2 null background (in Figure S2 legend for example). To claim so, they would need to compare the triple mutant with fat2(wa17);ftn-2(ok404) and show some rescue. However, we see in Figure 5H that ftn2(ok404) alone rescues fat-2(wa17). Thus, by comparing both figures, I see no additional effect of hif-1(et69) in an ftn-2(ok404) background. I actually think that this makes more sense, since the authors claim that hif-1(et69) is a gain-of-function mutation that acts through suppression of ftn-2 expression. Thus, I would expect that without ftn-2 from the beginning, hif-1(et69) does not have an additional effect, and this seems to be what we see from the data. Thus, I would suggest that the authors reformulate their claims regarding the effect of hif1(et69) in the ftn-2(ok404) background, which seems to be absent (consistently with what one would expect).

We completely agree with the reviewer and indeed this is the meaning that we tried to convey all along. The text has now been modified as follows:

"Lastly, *ftn-2(et68)* is still a potent *fat-2(wa17)* suppressor when hif-1 is knocked out (S2D Fig), suggesting that no other HIF-1-dependent functions are required as long as *ftn-2* is downregulated; this conclusion is supported by the observation that the potency of the ftn2(ok404) null allele to act as a *fat-2(wa17)* suppressor is not increased by including the *hif-1(et69)* allele (compare Fig 5H and S2E Fig)."